# Effect of Al$_2$O$_3$ and SiC Nano-Fillers on the Mechanical Properties of Carbon Fiber-Reinforced Epoxy Hybrid Composites

**S.M. Shahabaz, Prakhar Mehrotra, Hridayneel Kalita, Sathyashankara Sharma, Nithesh Naik** 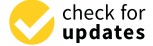**, Dilifa Jossley Noronha \* and Nagaraja Shetty \***

Department of Mechanical & Industrial Engineering, Manipal Institute of Technology, Manipal Academy of Higher Education, Manipal 576104, India
\* Correspondence: dj.noronha@manipal.edu (D.J.N.); nagaraj.shetty@manipal.edu (N.S.)

**Abstract:** Polymeric nanocomposites are an emerging research topic, as they improve fiber-reinforced composites' thermo-mechanical and tribological properties. Nanomaterials improve electrical and thermal conductivity and provide excellent wear and friction resistance to the polymer matrix material. In this research work, a systematic study was carried out to examine the tensile and hardness properties of a carbon fiber epoxy composite comprising nano-sized Al$_2$O$_3$ and SiC fillers. The study confirms that adding nano-fillers produces superior tensile and hardness properties for carbon fiber-reinforced polymer composites. The amount of filler loading ranged from 1, 1.5, 1.75, and 2% by weight of the resin for Al$_2$O$_3$ and 1, 1.25, 1.5, and 2% for SiC fillers. The maximum tensile strength gain of 29.54% and modulus gain of 2.42% were noted for Al$_2$O$_3$ filled composite at 1.75 wt.% filler loading. Likewise, enhanced strength gain of 25.75% and 1.17% in modulus gain was obtained for SiC-filled composite at 1.25 wt.% filler loading, respectively. The hardness property of nano-filled composites improved with a hardness number of 47 for nano-Al$_2$O$_3$ and 43 for nano-SiC, respectively, at the same filler loading.

**Keywords:** carbon fiber-reinforced polymer; tensile property; Barcol hardness; hybrid nano-composites

## 1. Introduction

Polymer matrices are reinforced with various types of fibers (carbon [1], glass [2,3], Kevlar [4], aramid [5], and natural fiber [6]) and different fillers (organic, inorganic, and metallic with micro-/nano-sized particulate fillers) to enable various applications and provide the maximum strength-to-weight ratio [7]. Research has shown that incorporating nanoparticles into epoxy nano-composites can significantly enhance the mechanical properties of the composite materials, including tensile strength, fracture toughness, impact, hardness, and fatigue properties [8–10]. However, polymer composites reinforced with different fillers are also part of recent trends in research these days. Low thermal expansion, adequate heat dissipation, and lightweight parts are needed in biomedical, structural, and aerospace applications [11,12]. Apart from the above applications, filler-bound composites are also used for fabricating sports equipment, household products, and commercial applications [13–15]. Therefore, different fillers are incorporated to enhance polymer composites' physical and mechanical properties. Blending or altering the components of epoxy mixtures makes it possible to manipulate their properties directly. Using plasticizers increases the polymer's elasticity and adjusts its glass transition temperature, while adding fillers enhances strength and imparts specific physical and chemical characteristics [16–19]. The effect of the filler on the polymer's properties is influenced by multiple factors, including the chemical nature of the polymer and filler, the filler's surface, particle size and shape, its ability to create structures, and the conformation of macromolecules and the polymer's structure [20,21]. As the filler content is varied, the properties of the polymer also change. Several researchers recently investigated micron-sized inorganic fillers' mechanical and

tribological properties with varying reinforcement fibers. According to De Cicco et al., adding various nanoparticles to the epoxy matrix has been found to alter its structure and improve its strength. This modification also enhances the bonding between the matrix and fibers, thus preventing the spread of microcracks during deformation and making the epoxy composite stronger [22]. Jiang et al. found that the char yield at 800 °C was improved from 14.3% to 26.2–26.6% when nano-$Al_2O_3$ was added [23]. Zhai et al. reported a significant increase in adhesion strength when 2 wt.% of nano-$Al_2O_3$ was added to the epoxy adhesive, compared to the pure epoxy composites [24]. According to Zheng et al., including $Al_2O_3$ nanoparticles increases impact strength, flexural modulus, and flexural strength by 84%, 29%, and 18%, respectively [25]. Wetzel et al. showed that adding $Al_2O_3$ nanoparticles into epoxy resin could improve stiffness, impact energy, and failure strain, even at low filler contents of 1–2 vol% [26]. Raffie et al. investigated tensile strength and fracture toughness of multi-walled carbon nanotubes (MWCNT) epoxy nano-composites. They observed a 31% increase in Young's modulus, a 40% increase in ultimate tensile strength, and a 53% increase in fracture toughness compared to neat epoxy composites [27]. Levy and Papazian investigated the tensile properties of aluminum matrix composites incorporated with SiC whiskers. The experimental results were compared with the finite element model results. They found that the analytical results were in good agreement with the experimental values. During their investigation, they observed a decrease in Young's modulus with an increase in SiC filler content [28]. Farzad and Saeedeh performed various numbers of both analytical and experimental investigations on graphene platelet-reinforced composites. In their study, they observed an increasing resistance to buckling response due to the presence of graphene platelets in the polymer matrix [29]. Also, in their other study, with a decrease in the plate thickness of graphene platelet-reinforced composites, critical buckling temperature was reduced [30]. Similarly, their further study of vibration analysis of graphene platelet-reinforced composites were experimented under thermal environments. They observed an increasing dimensionless frequency for graphene platelet-reinforced composite plates with an increase in weight fractions of graphene platelets [31,32]. Liang et al. found 62% and 76% increase in Young's modulus and tensile strength, respectively, by adding 0.7 wt.% of graphene oxide into the polymer matrix [33]. During the investigation by Parashar et al., 26% improved buckling stability was obtained with the 6% addition of graphene fillers in the polymer epoxy matrix composite [34]. Similarly, flexural strength and fracture toughness were improved by adding graphene platelets to alumina ceramic matrix composites. An increase of 30.75% and 27.20% in flexural strength and fracture toughness, respectively, was noted by Liu et al. [35].

Carbon fiber-reinforced polymer (CFRP) composites have led to a massive transition in the development of the structural, automotive, and aerospace sectors. They are extensively used due to their lessening of a component's weight in various industries without compromising its required strength [36]. CFRPs are also used as a load-bearing structural component in a high-temperature environment. Limited research has been conducted on incorporating nano-sized inorganic fillers with carbon fiber as a reinforcement material. The inclusion of fillers like SiC [37], $SiO_2$ [38], graphite [39], and $Al_2O_3$ [40] in woven glass fiber-reinforced epoxy composites have been investigated, and an increasing trend in mechanical properties with filler content has been found [41,42]. Hussain et al. examined the effect of alumina particle size (1 µm and 25 nm) on the mechanical properties of carbon-epoxy composites [43]. They discovered that hybridizing carbon fiber-reinforced epoxy composites with $Al_2O_3$ micro-/nano-fillers significantly improved mechanical properties. Ozsoy et al. found maximum flexural strength when adding 4 wt.% $Al_2O_3$ nano-fillers to the polymer matrix [44].

Similarly, with the implementation of the ultrasonic dispersion technique to disperse $Al_2O_3$ nano-fillers into the epoxy matrix, with varying weight percentages (1–5 wt.%), the toughness of composites was determined with carbon fiber as reinforced material. They reported the maximum improvement in toughness was noted for 2 wt.% $Al_2O_3$ filled composite [45]. Megahed et al. investigated the effect of nano silica fillers incorporated

into glass fiber epoxy composite to determine the hardness and wear resistance of the composite. They observed a 50.07% increase in hardness at 1 wt.% filler loading, and wear resistance was improved at 0.5 wt.% filler loading compared to the unfilled glass fiber epoxy composites [46]. Similarly, Chisholm studied the effect of SiC micro- and nano-fillers incorporated into the carbon fiber epoxy composite fabricated using vacuum-assisted resin transfer molding at 1.5 and 3 wt.% filler loading. They observed increased flexural strength at 1.5 wt.% of nano-filler loading [47].

The previous studies show that as filler content increases, mechanical property increases, and after a particular addition, the property quickly deteriorates. This phenomenon can be explained based on the interaction between nano-fillers and polymeric chains of the matrix. The stronger the particle–particle interaction between fillers and polymeric chains of epoxy, the more mechanical properties tend to improve. However, with the converse to the above statement, as the filler loading increases above an optimum level, the reverse action occurs, causing the nano-filler localization (agglomeration) in the polymer matrix and decreasing mechanical properties.

In addition to the above explanations, the continued high-speed mixing of filler and epoxy solution for a longer duration lessens the formation of nano-filler agglomerations. As a result, the gaps are reduced, and a high degree of interaction occurs between the filler and polymer chains. Due to this, stronger hydrogen bonding is formed between the filler epoxy phases, improving mechanical and thermal properties. Also, the external factor changing the behavior of mechanical properties is the appropriate particle size (micro/nano). As seen in the above literature, improved mechanical properties were noted for nano-filled composite; that is, the smaller the particle size, the larger the surface area, resulting in better dispersion within macromolecules, and hence improving the mechanical properties of composites [48,49].

Literature surveys have shown limited research focusing on the impact of incorporating $Al_2O_3$ and SiC into carbon fiber-reinforced epoxy composites. As a result, it would be of great interest to examine the effect of modifying these composites with $Al_2O_3$ and SiC nano-fillers and gain a deeper understanding of their tensile and hardness behavior. In the present investigation, two types of nano-fillers ($Al_2O_3$ and SiC) infused with epoxy resin (polymer matrix) through the mixing combination of ultrasonication and magnetic stirring technique were developed with carbon fiber as the reinforcement material. This research work aims to obtain the exact filler loading of the above nano-fillers to achieve maximum tensile and hardness properties of fabricated composites.

## 2. Fabrication of CFRP and Hybrid Nano-Composites

The unfilled CFRP and hybrid CFRP nano-composites were manufactured using the hand lay-up method followed by compression molding. Sixteen layers of uni-directional (UD) carbon fabric provided by Bhor Chemicals & Plastics Pvt. Ltd., Maharashtra, India, were laid in a quasi-isotropic fiber orientation of $(0/\pm45/90°)$. The room temperature curing was performed for 24 h, and the composites' thickness was maintained at $3 \pm 0.2$. Bisphenol-A epoxy resin and amine-based hardener were used to fabricate composites obtained from Bhor Chemicals & Plastics Pvt. Ltd. The resin and hardener mixing ratio of 100:30 was performed, as supplied by the manufacturer. The two types of nano-fillers used for improving the CFRP composites were alumina ($Al_2O_3$) and silicon carbide (SiC) with average diameter sizes of 25 and 50 nm, respectively, obtained from Sisco Research Laboratories Pvt. Ltd., Mumbai, India. The physical properties of carbon fabric, resin, and nano-fillers are represented in Table 1.

**Table 1.** Properties of materials used for fabrication of CFRP and hybrid nano-composites.

| Material | Density (g/cm$^3$) | Viscosity (mPa) | Size | Color |
|---|---|---|---|---|
| UD carbon fibre | 1.8 | - | 7 μm | Black |
| Epoxy resin | 1.2 | 11,000 | - | Transparent |
| Hardener | 0.95 | 50 | - | Transparent |
| $Al_2O_3$ | 3.9 | - | 20–30 nm | White |
| SiC | 3.2 | - | 50 nm | Grey |

UD—uni-directional, $Al_2O_3$—alumina, SiC—silicon carbide.

The uniform dispersion of nano-fillers into resin was achieved using a 2 KW high-power probe sonicator with a 25 mm probe diameter, followed by the magnetic stirring method. The sonication parameters selected for the study were 15 s on and 30 s off for 1 h with a 50% amplitude, respectively. The nano-filler resin solution was immersed in a water bath to avoid overheating and damping the cavitation process. The dispersion of nano particles is challenging due to the dense nature of the epoxy resin because increasing the weight fraction of these nano particles in epoxy increases the viscosity of epoxy. Also, while performing the sonication process, an increase in temperature is observed, causing deterioration of mechanical properties of the epoxy solution. To avoid this, the sonication process was performed in a pulsed mode which hinders the temperature increase rate, allowing for better temperature control. For the present work, the sonication of nano particles was carried out in a pulsed mode with 15 s on and 30 s off, for 60 min duration at amplitude setting of 50%.Furthermore, the nano-filler solution was stirred for 30 min using a magnetic stirrer at a 1000 rpm rotational speed. The uniform dispersion of the nano-filler solution was determined using a field-emission scanning electron microscope (FE-SEM). Table 2 represents the designation used for each composite and the different filler and epoxy loadings in 50 wt.% of carbon fabric reinforcement.

**Table 2.** Composites fabricated at different weight percentages.

| Composites | Fiber (wt.%) | Epoxy Resin (wt.%) | Nano-Filler Loading (wt.%) |
|---|---|---|---|
| CFRP (neat composite) | 50 | 50 | - |
| 1 wt.% $Al_2O_3$ | 50 | 49 | 1 |
| 1.5 wt.% $Al_2O_3$ | 50 | 48.5 | 1.5 |
| 1.75 wt.% $Al_2O_3$ | 50 | 48.25 | 1.75 |
| 2 wt.% $Al_2O_3$ | 50 | 48 | 2 |
| 1 wt.% SiC | 50 | 49 | 1 |
| 1.25 wt.% SiC | 50 | 48.75 | 1.25 |
| 1.5 wt.% SiC | 50 | 48.5 | 1.5 |
| 2 wt.% SiC | 50 | 48 | 2 |

CFRP—carbon fiber-reinforced polymer, $Al_2O_3$—alumina, SiC—silicon carbide.

## 2.1. Tensile Test

Fabricated composites were cut based on ASTM D-638 using an abrasive water jet cutting machine to reduce the damage during cutting. A tensile test was performed on the universal testing machine (Instron 3366) with a displacement rate of 1 mm/min. Dumbbell-shaped specimens were cut with a gauge length of 7.62 mm and a width of 3.18 mm, as represented in Figure 1a. At least five samples were tested for unfilled and nano-filled composite, and their average value was noted. The ultimate tensile strength was evaluated for the maximum load before the failure of the composites. Figure 1b represents the specimen's fracture surface. Based on ASTM standards, the tensile strength and modulus were calculated from the following Equations (1)–(3), as mentioned below.

$$\text{Tensile strength, } \sigma = \frac{P_{max}}{A_o} \text{ (MPa)} \tag{1}$$

$$\text{Strain, } \varepsilon = \frac{\Delta L}{L_o} \qquad (2)$$

$$\text{Tensile modulus, } E = \frac{\sigma}{\varepsilon} \qquad (3)$$

where $P_{max}$—maximum load at the failure (N), $A_o$ = original cross-sectional area (mm$^2$) = b × h, b—specimen width (mm), h—specimen thickness (mm), $\Delta L$—Elongation (mm), $L_o$ = original gauge length (mm).

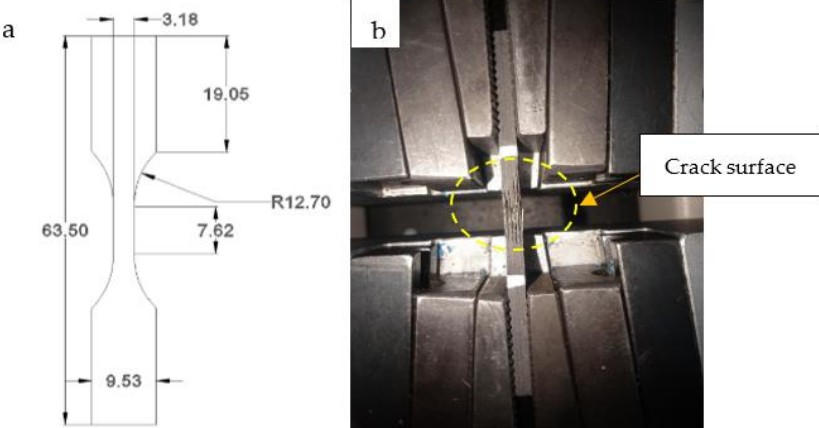

**Figure 1.** Tensile specimen of composite: (**a**) geometry of tensile specimen; (**b**) tensile failure specimen.

### 2.2. Hardness Test

A hardness test was performed based on ASTM D2583 employing a Barcol hardness tester. Five specimens of 30 × 10 mm$^2$ were cut with abrasive water jet cutting at five different positions of each composition, as shown in Figure 2. The distance of 5 mm was maintained for each indentation as per standards.

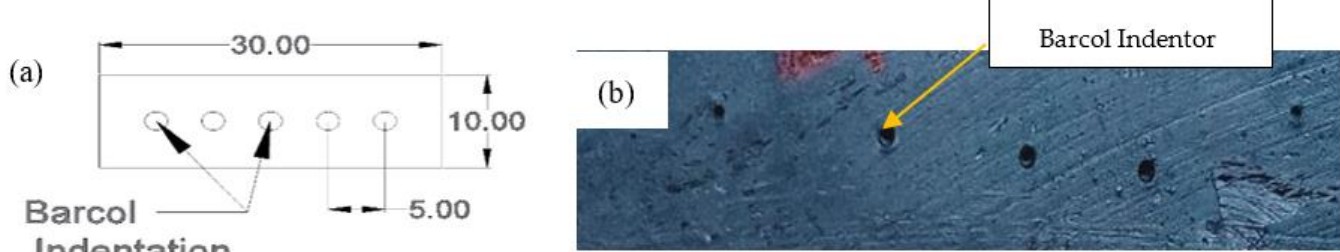

**Figure 2.** Hardness specimen of composite: (**a**) geometry of hardness specimen; (**b**) indentations made using Barcol hardness tester.

### 3. Results and Discussions

#### 3.1. Analysis of Nano-Fillers Using FESEM

Figure 3 represents the uniform distribution of nano-fillers analyzed using a field emission scanning electron microscope at 1.75 wt.% Al$_2$O$_3$ and 1.25 wt.% SiC weight fractions. The SEM images verify that the sonication and magnetic stirring successfully improve the uniform dispersion of nano-fillers into the resin solution. As reported by Kaybal et al., an increase in the weight fraction of nano-fillers leads to their agglomeration due to Van der Waals attractive forces, resulting in the formation of clusters, as observed in Figure 4, which acts as a damaging effect on the mechanical properties of the nano-composites [40].

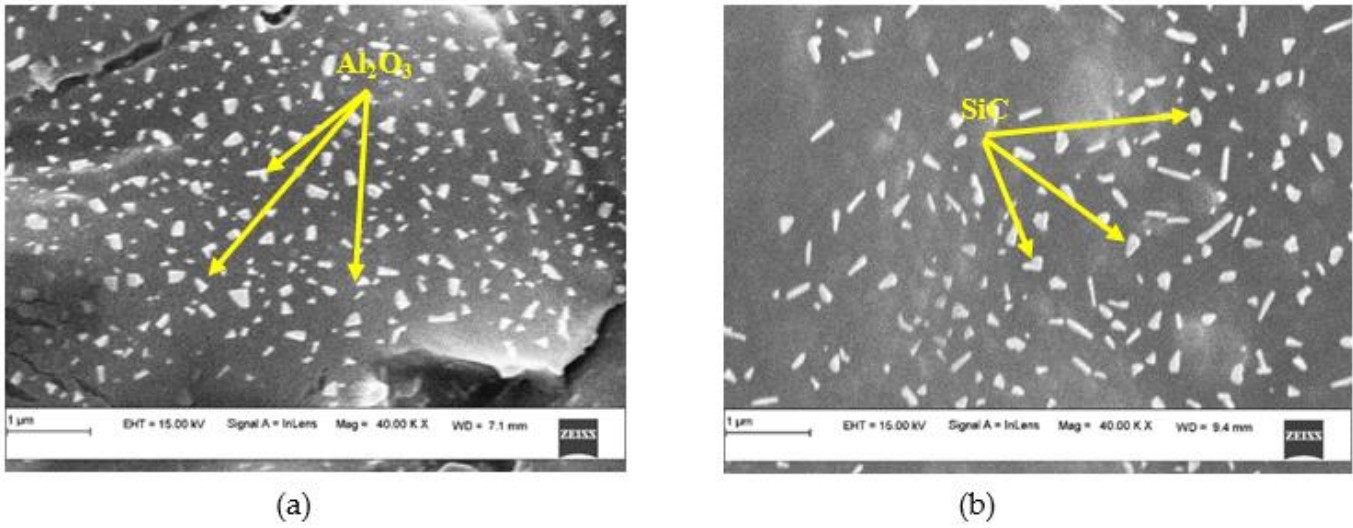

**Figure 3.** Distribution of nano-fillers: (**a**) 1.75 wt.% $Al_2O_3$; (**b**) 1.25 wt.% SiC [50].

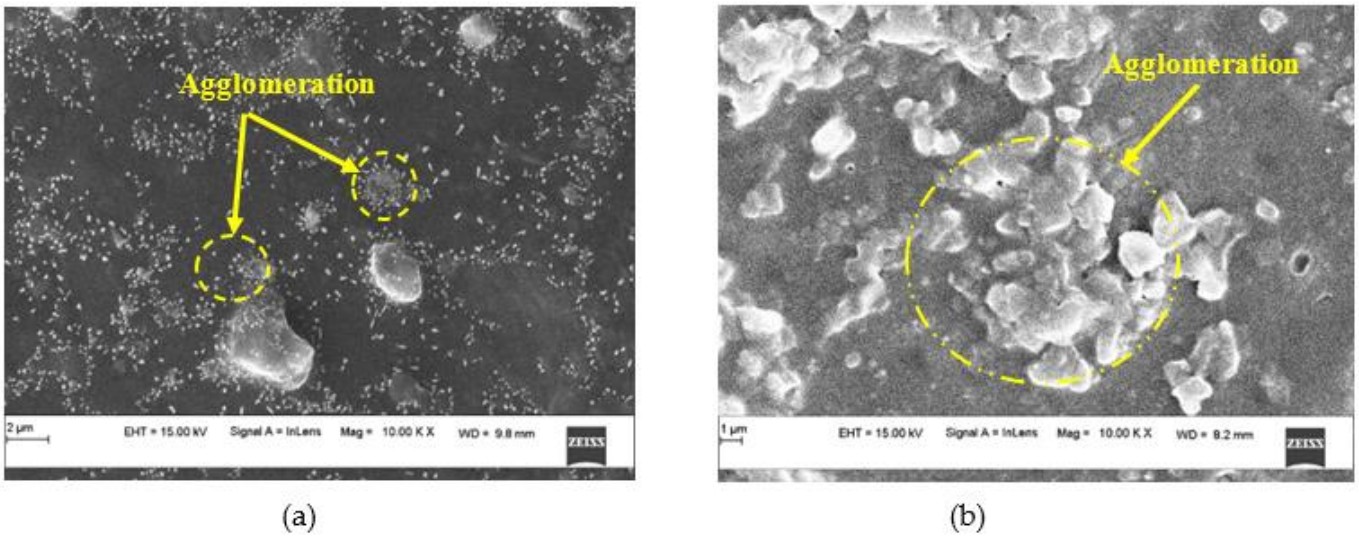

**Figure 4.** Agglomeration of nano-fillers: (**a**) 2 wt.% $Al_2O_3$; (**b**) 2 wt.% SiC [50].

### 3.2. Tensile Behavior of CFRP and Hybrid Nano-Composites

The effect of filler loading on tensile strength and modulus is represented in bar charts for $Al_2O_3$ and SiC hybrid nano-composites. It is observed that, with the increasing filler content, the tensile strength increases and then lowers. The reduction in tensile property indicates the agglomeration of nano-fillers after an optimum filler loading. Among the two filled composites, the maximum tensile strength was noted for $Al_2O_3$ hybrid nano-composites (1.75 wt.%) followed by SiC hybrid nano-composites (1.25 wt.%) and unfilled composite. Tensile strength improved from 10.34, 22.66, and 29.54% for $Al_2O_3$ filled composite at 1, 1.5, and 1.75 wt.% filler loading, respectively, whereas, for SiC filler loading, increasing strength of 14.22 and 25.75% was noted at 1 and 1.25 wt.% filler loading, respectively. The increase in tensile strength of nano-filled composites is higher than the unfilled composite due to the high interfacial bonding and wetting between the nano-filler and polymer matrix as observed in Section 3.1.

In addition to the above explanation, the higher strength achieved is due to the oxygen atom in $Al_2O_3$ fillers that builds adequate hydrogen bonding between polymeric chains of epoxy resin and nano-fillers. The even distribution of $Al_2O_3$ nano-fillers in the epoxy matrix decreases the mobility of epoxy chains, creating highly immobile nanolayers around each particle. This confinement of non-contact matrix chains leads to a more complex

network, as the polar $Al_2O_3$ nano-fillers fill the spaces between chains and attract resin molecules during curing [51]. The oxygen in $Al_2O_3$ also creates strong hydrogen bonds with the polymeric chains, further increasing the constraints between particles and chains, as represented in Figure 5 [52]. These extra forces enhance the nano-composites' strength compared to non-filled composites. The results obtained from the tensile test are illustrated in Tables 3 and 4. The bar graph for tensile strength and modulus at different filler loading for hybrid $Al_2O_3$ nano-composites are represented in Figures 6 and 7, and for hybrid SiC nano-composites in Figures 8 and 9.

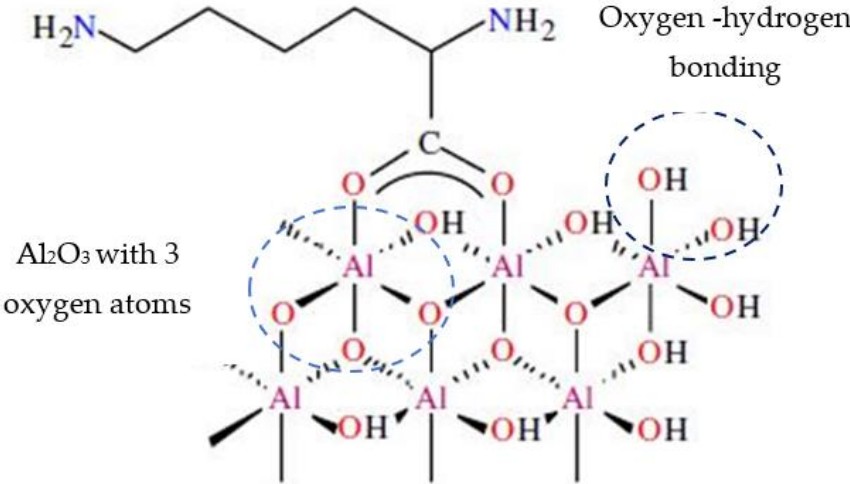

**Figure 5.** Epoxy resin chain reaction with $Al_2O_3$ nano-fillers [52].

**Table 3.** Tensile behavior of CFRP and hybrid $Al_2O_3$ nano-composites.

| Material | Tensile Strength (MPa) | | Strength Gain in (%) | Tensile Modulus (GPa) | | Modulus Gain in (%) |
|---|---|---|---|---|---|---|
| | Average | Std. Dev. | | Average | Std. Dev. | |
| CFRP | 233 | 14.96 | - | 148 | 2.12 | - |
| 1 wt.% $Al_2O_3$ | 257 | 14.05 | 10.34 | 149 | 3.41 | 0.67 |
| 1.5 wt.% $Al_2O_3$ | 286 | 9.92 | 22.66 | 150 | 2.16 | 1.28 |
| 1.75 wt.% $Al_2O_3$ | 302 | 18.10 | 29.54 | 152 | 2.40 | 2.42 |
| 2 wt.% $Al_2O_3$ | 279 | 4.46 | 19.65 | 149 | 3.23 | 1.13 |

**Table 4.** Tensile properties of CFRP and hybrid SiC nano-composites.

| Material | Tensile Strength (MPa) | | Strength Gain in (%) | Tensile Modulus (GPa) | | Modulus Gain in (%) |
|---|---|---|---|---|---|---|
| | Average | Std. Dev. | | Average | Std. Dev. | |
| CFRP | 233 | 14.96 | - | 148 | 1.34 | - |
| 1 wt.% SiC | 266 | 11.85 | 14.22 | 149 | 2.35 | 1.03 |
| 1.25 wt.% SiC | 293 | 10.34 | 25.75 | 150 | 3.40 | 1.17 |
| 1.5 wt.% SiC | 268 | 13.35 | 15.10 | 149 | 4.50 | 1.03 |
| 2 wt.% SiC | 240 | 18.03 | 3.19 | 148 | 2.10 | 0.22 |

In the case of hybrid SiC nano-composite at 1.25 wt.% filler loading, the maximum tensile properties are noted, as shown in Table 4. Their respective bar graphs are represented in Figures 8 and 9. However, the increase in strength is higher than the unfilled composite but less than the $Al_2O_3$ hybrid nano-composites. As explained in the above section about the bonding behavior of oxygen atoms of $Al_2O_3$ fillers with polymer matrix, in the case of SiC fillers, similar bonding occurs between the silica and oxygen atoms of the polymer matrix. SiC particles bond with epoxy resin through the interaction between silica and the oxygen atom of epoxy and carbon with the hydrogen atom of epoxy. However, the lack

of oxygen–hydrogen bonding, which is more vital than carbon–hydrogen bonding, leads to a decrease in the load-bearing capacity of the composite and a reduction in strength compared to hybrid $Al_2O_3$ nano-composites [53]. In addition to the above explanation, another cause of the decrease in strength is the lower weight of $Al_2O_3$ nano-fillers compared to SiC nano-fillers. The higher weight of SiC nano-fillers tends to settle down after mixing, causing agglomeration of fillers [50]. Several researchers noted this similar behavior during the fabrication of nano-composites without any fabric as a reinforcement component.

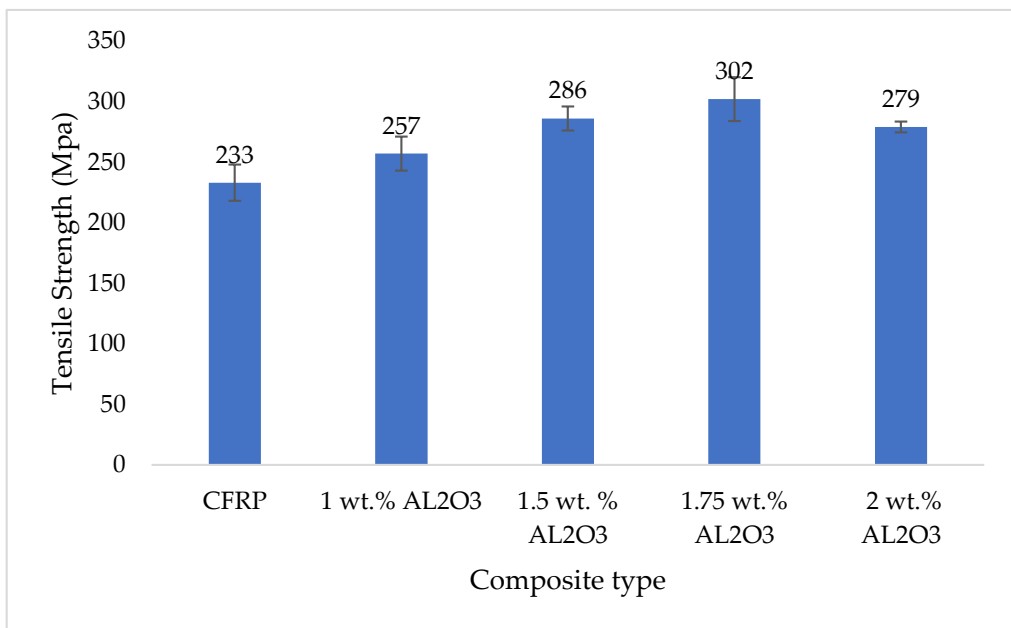

**Figure 6.** Tensile strength of CFRP and hybrid $Al_2O_3$ nano-composites.

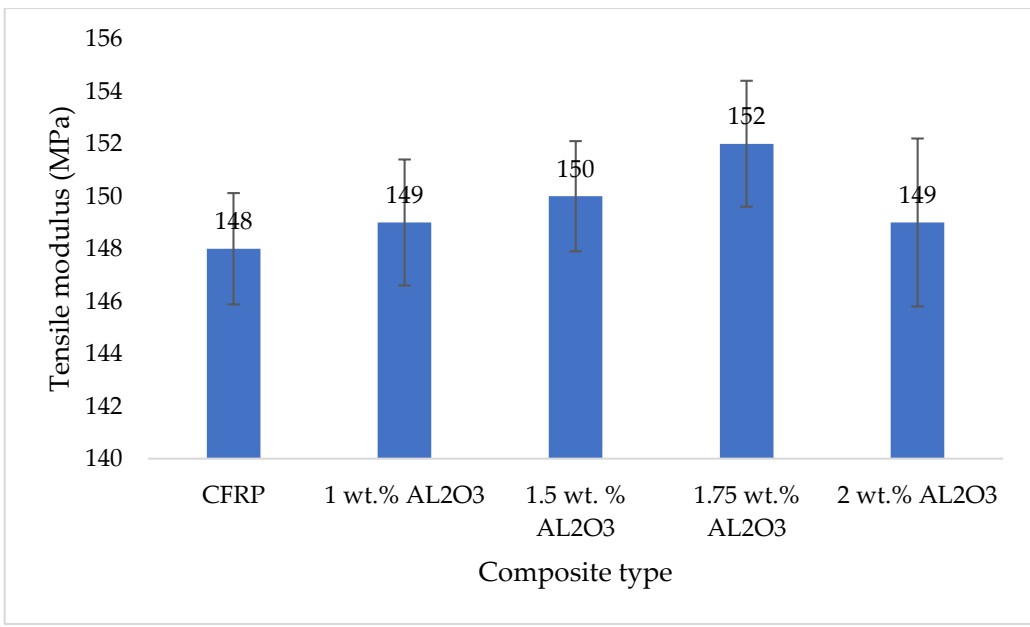

**Figure 7.** Tensile modulus of CFRP and hybrid $Al_2O_3$ nano-composites.

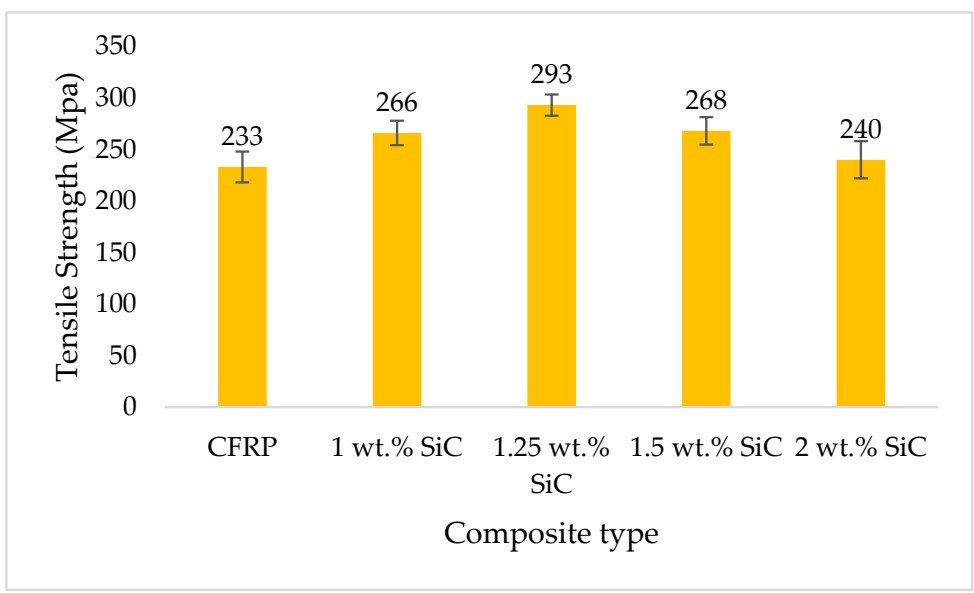

**Figure 8.** Tensile strength of CFRP and hybrid SiC nano-composites.

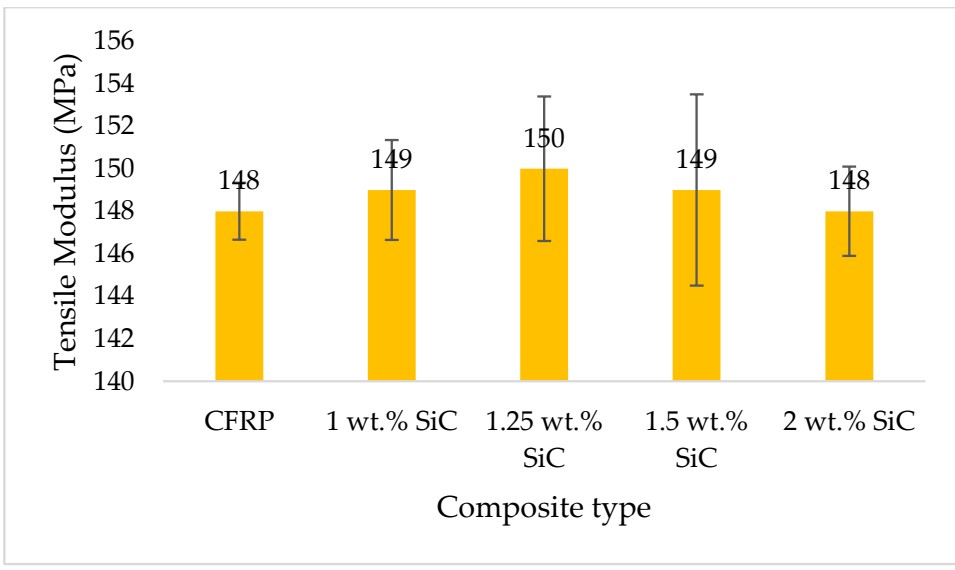

**Figure 9.** Tensile modulus of CFRP and hybrid SiC nano-composites.

*3.3. Characterization of Tensile Failed Specimens*

The fracture behavior of tensile failed specimens was examined using a scanning electron microscope (SEM). The fractured surfaces of hybrid $Al_2O_3$ nano-composites (1.75 wt.%), hybrid SiC nano-composites (1.25 wt.%), and unfilled composites are shown in Figures 10–12, respectively. For nano-filled composites, it is seen that the nano-fillers have been uniformly distributed, displaying strong bonding and wetting of particles with the polymer matrix. The fiber orientation enhances nano-composite strength along with the uniform dispersion of nano-fillers. The SEM images show that fiber pull-out, debonding, and fiber fracture are the different failure mechanisms observed.

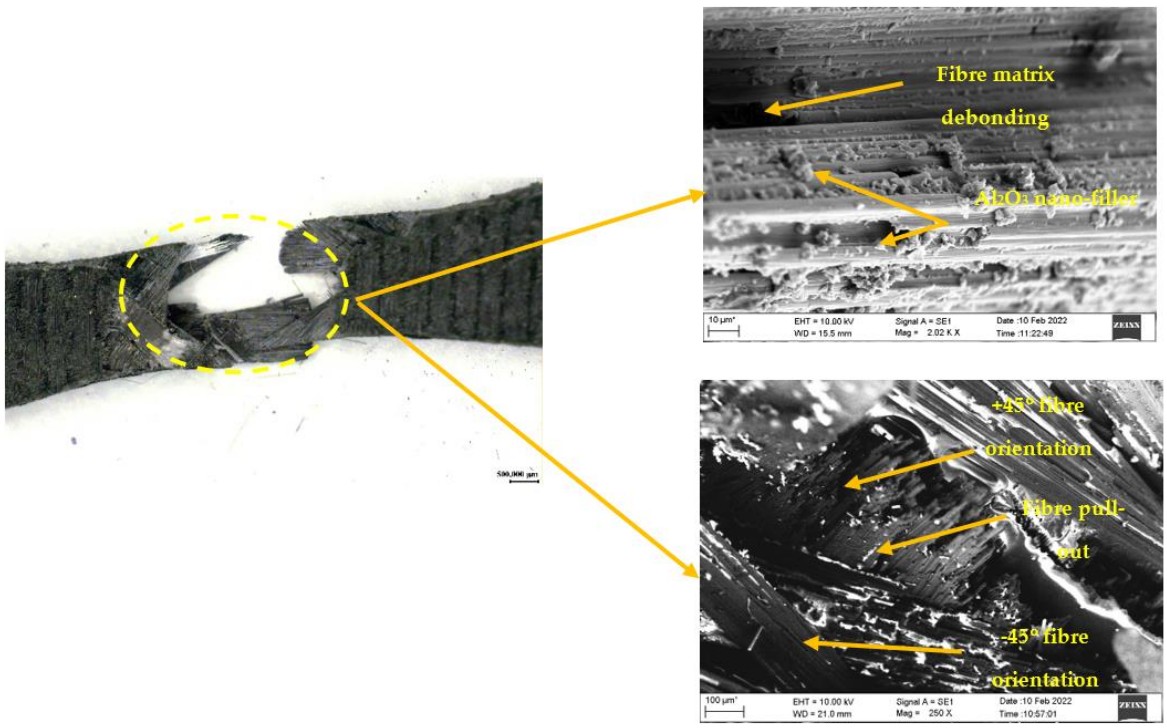

**Figure 10.** SEM images of tensile failure specimens of hybrid $Al_2O_3$ nano-composites at 1.75 wt.%.

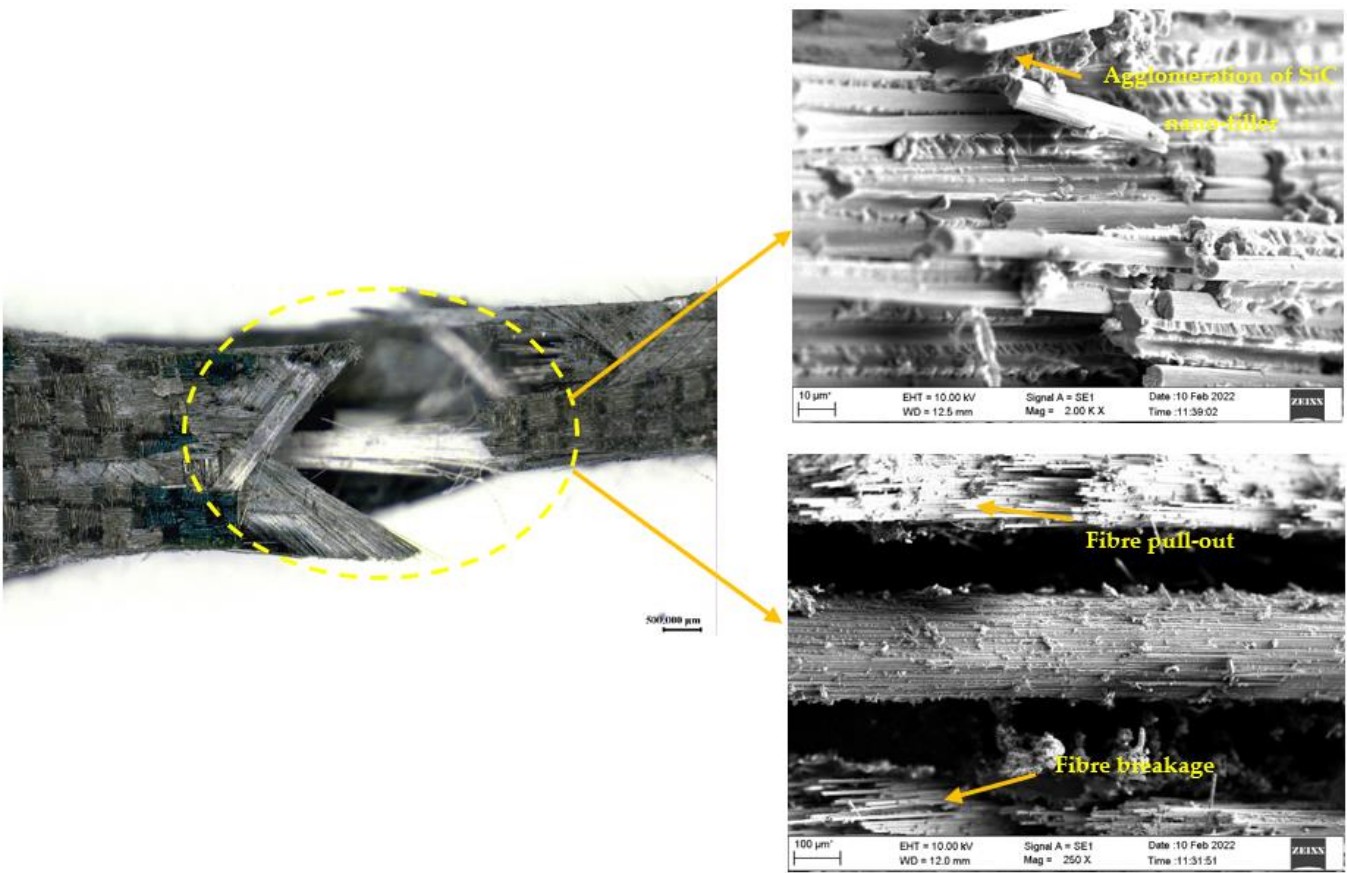

**Figure 11.** SEM images of tensile failure specimens of hybrid SiC nano-composites at 1.25 wt.%.

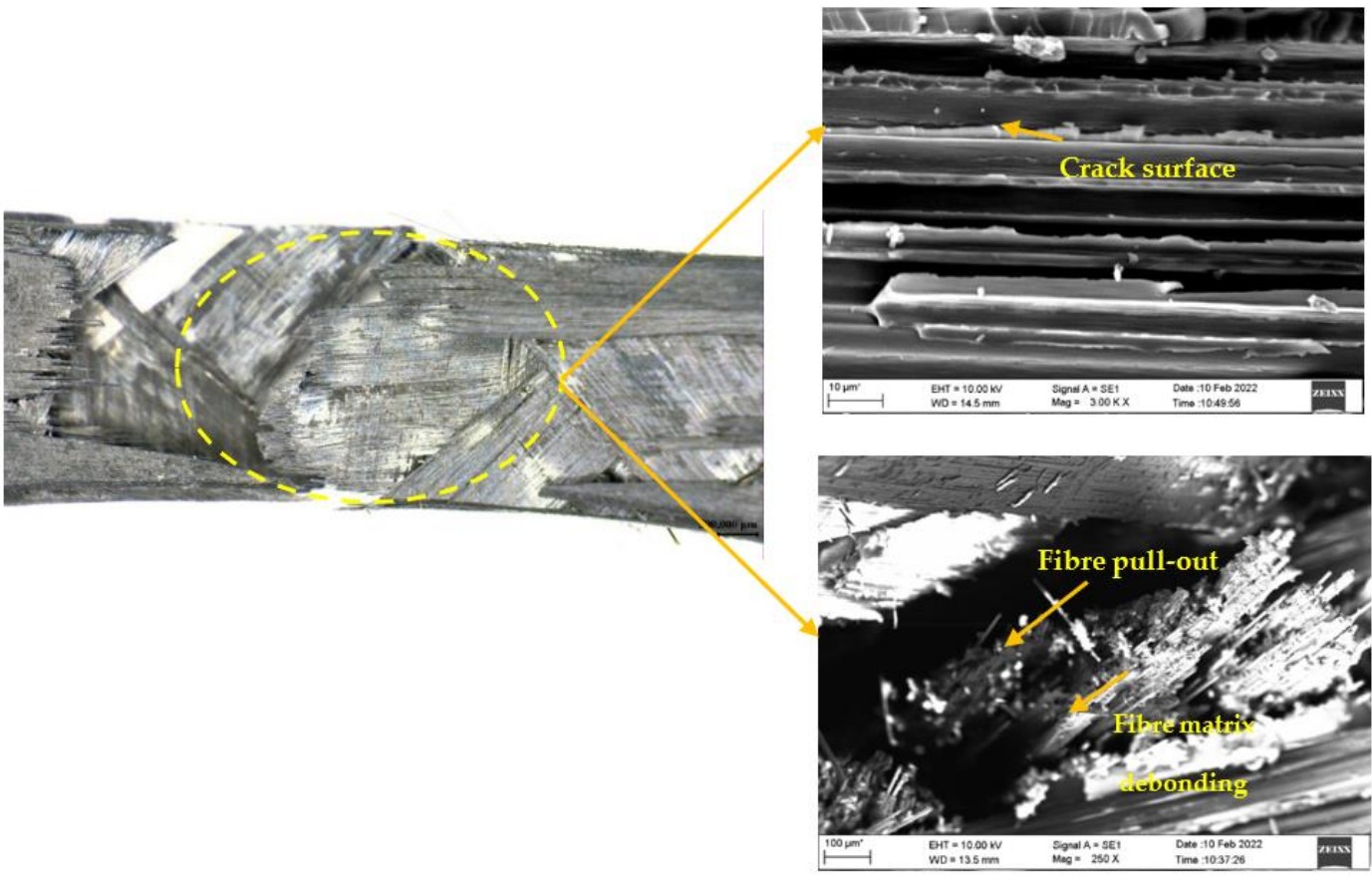

**Figure 12.** SEM images of tensile failure specimens of unfilled CFRP composite.

### 3.4. Hardness Behavior of CFRP and Hybrid Nano-Composites

One of the most significant factors that affect a material's wear resistance is its hardness. The Barcol hardness for the unfilled and nano-filled composites was measured and noted using a Barcol hardness tester. The Barcol hardness number is represented using a bar graph in Figures 13 and 14. The hardness number improved from 37, 43, and 47 for $Al_2O_3$-filled composite at 1, 1.5, and 1.75 wt.% filler loading, respectively, whereas, for SiC filler composite, increasing hardness numbers of 32 and 43 were noted at 1 and 1.25 wt.%, respectively, as shown in Table 5. The findings indicate that adding filler enhances the hardness of the nano-filled composites. It is concluded that the hardness increases as the filler particles form a closer packing of atoms with the increasing filler content, allowing shorter bond length; hence, hardness increases. The maximum hardness value was noted for $Al_2O_3$ hybrid nano-composites at 1.75 wt.% filler loading with a slight hardness decrease for SiC hybrid nano-composites at 1.25 wt.%. The minimum hardness was observed for unfilled composite with a hardness value of 27.

**Table 5.** Hardness number of CFRP and hybrid nano-composites.

| Material | Hardness Number | | Material | Hardness Number | |
|---|---|---|---|---|---|
| | Average | Std. Dev. | | Average | Std. Dev. |
| CFRP | 27 | 3.78 | CFRP | 27 | 3.78 |
| 1 wt.% $Al_2O_3$ | 39 | 2.17 | 1 wt.% SiC | 32 | 2.55 |
| 1.5 wt.% $Al_2O_3$ | 43 | 2.30 | 1.25 wt.% SiC | 43 | 3.61 |
| 1.75 wt.% $Al_2O_3$ | 47 | 1.82 | 1.5 wt.% SiC | 37 | 1.95 |
| 2 wt.% $Al_2O_3$ | 36 | 2.70 | 2 wt.% SiC | 33 | 1.71 |

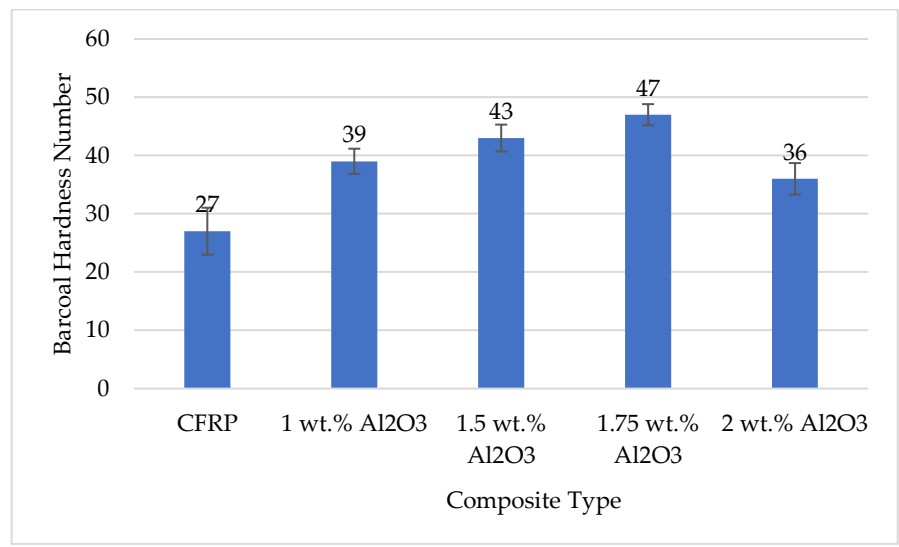

**Figure 13.** Barcol hardness number of CFRP and Al₂O₃ hybrid nano-composites.

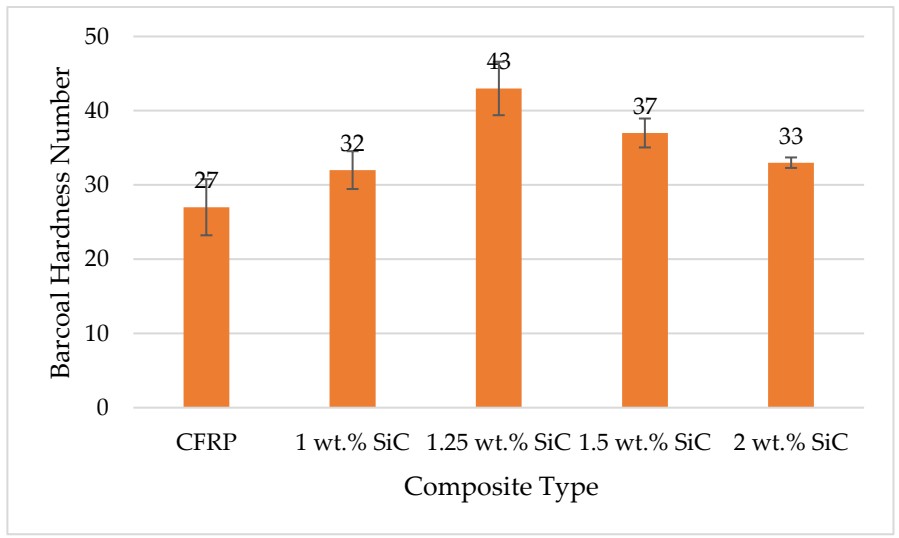

**Figure 14.** Barcol hardness number of CFRP and SiC hybrid nano-composites.

## 4. Conclusions

The present study reveals the influence of nano-fillers (Al$_2$O$_3$ and SiC) in enhancing CFRP hybrid nano-composites' tensile and hardness properties. The primary outcome of the research was to determine the exact optimum filler loading of the nano-fillers, as mentioned above, at which the property increases. The following conclusions are drawn from the experimental study.

1. The maximum tensile and hardness properties were noted for Al$_2$O$_3$ hybrid nano-composites at filler loading of 1.75 wt.%, for SiC hybrid nano-composites at 1.25 wt.% filler loading, in comparison to the unfilled composite.
2. The ultrasonication technique followed by the magnetic stirring method was an effective method for dispersing the nano-fillers into the polymer matrix.
3. The optimum range of filler loading obtained is 1.75 wt.% for Al$_2$O$_3$ and 1.25 wt.% for SiC nano-fillers for higher mechanical properties.
4. The superior mechanical properties obtained for Al$_2$O$_3$ hybrid nano-composites represent the importance of solid bond formation, of oxygen-hydrogen bonding between Al$_2$O$_3$ and the epoxy polymer matrix.

5. The lower mechanical properties for SiC hybrid nano-composites indicate the lower bond capacity of carbon–hydrogen bonding as compared to the oxygen–hydrogen bonding of $Al_2O_3$ and the polymer matrix.

**Author Contributions:** Conceptualization, investigation, methodology, writing—original draft, S.M.S.; conceptualization, writing—review and editing, P.M.; conceptualization, writing—review and editing, H.K.; conceptualization, methodology, writing—review and editing, project administration, S.S.; supervision, project administration, N.N.; supervision, project administration, D.J.N.; conceptualization, methodology, writing—review and editing, project administration, N.S. All authors have read and agreed to the published version of the manuscript.

**Funding:** This research received no external funding.

**Data Availability Statement:** Not applicable.

**Conflicts of Interest:** The authors declare no conflict of interest.

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
