# Peer review of "Effect of Al2O3 and SiC Nano-Fillers on the Mechanical Properties of Carbon Fiber-Reinforced Epoxy Hybrid Composites"

_jcs, doi:10.3390/jcs7040133_

Round 1
Reviewer 1 Report
The manuscript under the title: “Influence of Al2O3 and SiC nano fillers on tensile and hardness behavior of carbon fibre epoxy composite” is in line with Journal of Composites Science. This topic is relevant and will be of interest to the readers of the journal. It based on original research. This research has scientific novelty and practical significance. The article has a typical organization for research articles.
Before the publication it requires significant improvements, especially:
- The "Introduction" section: it has been proven that the effect of various modifying additives and fillers on the physico-chemical and mechanical properties of epoxy polymer composites is determined by many factors: ……. I think the related references should be cited corresponding to each aspect, e.g. (but not limited to these), which will undoubtedly improve the "Introduction" section:
Polymers 2021, 13(15), 2421; https://doi.org/10.3390/polym13152421
- Polymers 2022, 14(22), 4852; https://doi.org/10.3390/polym14224852
- Polymers 2022, 14(2), 338; https://doi.org/10.3390/polym14020338
- Polymers 2022, 14(9), 1841; https://doi.org/10.3390/polym14091841
- Russ J Appl Chem 86, 765–771 (2013). https://doi.org/10.1134/S107042721305025X
- Polymers 2022, 14(7), 1325; https://doi.org/10.3390/polym14071325
2. The "Introduction" section: show what is the scientific novelty of your research and how it differs from those described in the literature.
4 3.What justifies the choice of pressing pressure (2 kPa) in the manufacture of reinforced composites?
5 4. In table.1. it is necessary to add more detailed information (viscosity, epoxy group content, etc.) about the epoxy resin and the hardener used.
6 5. Line 176-179. Talking about increasing the modulus of elasticity is not at all appropriate, because the values of the modulus of elasticity for all compositions change within the experimental error.
7 6. Line 182-187 and Line 194-203. These conclusions must be proven or referenced in the literature (reference [23] is your self-citation and is not suitable for proof).
8 7. Table 3-4, Figure 3-6. does it make sense to give numbers with an accuracy of hundredths (0.00), if you have an experimental error of 5-6%, I suggest rounding all numbers to whole values.
9 8. The mechanical properties which are significantly showing the efficacy of the modification, should be stressed more, in particular by showing some stress/strain graphic comparison, in order to determine also the elastic and plastic behavior modifications.
1 9. Conclusion 2 (line 275-276) in this article you did not consider various options for homogenizing epoxy compositions, so this conclusion is inappropriate.
Author Response
Response (revision notes) to the Reviewer’s Comments:
The authors of this paper would like to sincerely thank the reviewer for their valuable comments/suggestions made. Please find the response for each comment in the following paragraphs.
- The "Introduction" section: it has been proven that the effect of various modifying additives and fillers on the physico-chemical and mechanical properties of epoxy polymer composites is determined by many factors: ……. I think the related references should be cited corresponding to each aspect, e.g. (but not limited to these), which will undoubtedly improve the "Introduction" section:
Response: As per the reviewer’s suggestion all the references as mentioned by reviewer has been added and cited in the Introduction section.
- The "Introduction" section: show what is the scientific novelty of your research and how it differs from those described in the literature.
Response: As per the reviewer’s suggestion, the novelty of our research work has been added in the end of introduction section.
- What justifies the choice of pressing pressure (2 kPa) in the manufacture of reinforced composites?
Response: As per the reviewer’s suggestion, the pressing pressure (2kPa) has been removed in the revised manuscript.
- In table.1. it is necessary to add more detailed information (viscosity, epoxy group content, etc.) about the epoxy resin and the hardener used.
Response: As per the reviewer’s suggestion, In the revised manuscript, more information about epoxy and hardener has been added in Table 1 as well as in section 2 (Line 120).
- Line 176-179. Talking about increasing the modulus of elasticity is not at all appropriate, because the values of the modulus of elasticity for all compositions change within the experimental error.
Response: As per the reviewer’s suggestion, the increasing modulus of elasticity sentence has been removed in the revised manuscript.
- Line 182-187 and Line 194-203. These conclusions must be proven or referenced in the literature (reference [23] is your self-citation and is not suitable for proof).
Response: The conclusions are proven with references in section 3.1 and 3.2.
- Table 3-4, Figure 3-6. does it make sense to give numbers with an accuracy of hundredths (0.00), if you have an experimental error of 5-6%, I suggest rounding all numbers to whole values.
Response: As per the reviewer’s suggestion, the result numbers have been rounded off to whole values.
- The mechanical properties which are significantly showing the efficacy of the modification, should be stressed more, in particular by showing some stress/strain graphic comparison, in order to determine also the elastic and plastic behavior modifications.
Response: We respect and acknowledge the reviewer’s suggestion, but due to some circumstances we lost the file which had the stress strain data, that is why we were not able to add stress strain graphs at the first time only (that is at the beginning submission of the manuscript). We hope you, understand our compulsion.
- Conclusion (line 275-276) in this article you did not consider various options for homogenizing epoxy compositions, so this conclusion is inappropriate.
Response: We have added and explained, in the revised manuscript about the homogenizing epoxy compositions in section 3.1.
Author Response
Response (revision notes) to the Reviewer’s Comments:
The authors of this paper would like to sincerely thank the reviewer for their valuable comments/suggestions made. Please find the response for each comment in the following paragraphs.
- What is the distribution pattern of the fillers in the composite layer? Is it random? or Does it follow any model? It is necessary to discuss it as one of the effectful parameters on the mechanical properties. It is recommended to present different patterns and distribution models to carry out the best mechanical properties.
Response: As per the reviewer’s suggestion, distribution of fillers in epoxy solution has been explained in section 3.1, in the revised manuscript.
- How are the authors justifying the tensile strength growing by the act of interfacial bonding and wetting in terms of physical attitude? I’m not satisfied with the explanations reported in the manuscript.
Response: As per the reviewer’s suggestion, authors have added and explained with references in section 3.1 and section 3.2, about the increasing in tensile strength of nano-composites.
- It is better to present a more comprehensive title using some other words and phrases. This title seems so elementary.
Response: As per the reviewer’s suggestion, authors have reframed the title.
- Authors are recommended to check their English writing and improve the text. There are several grammar errors and issues. The authors may need to check the work carefully to correct them all.
Response: As per the reviewer’s suggestion, the authors have checked and corrected several grammar errors and issues in the revised manuscript.
- Table 2 is not well designed. It is suggested to change the composite names column and make a better category and naming.
Response: As per the reviewer’s suggestion, authors have changed the category and naming.
- Authors should try to set the captions again in order to align and justify all in the same format. (For instance, look at Figure 8 and Figure 9)
Response: As per the reviewer’s suggestion, authors have set and aligned the captions for all the 3 Figures, in the revised manuscript.
- The introduction section is weak and should be strongly rewritten again. Authors should try to bring a comprehensive literature review of the related works of reinforced composite structures which in the last decades were attracting much more attention from scientists and industrial companies as well. There is a wide literature review in this area. Some of the suggested articles are listed below.
Response: As per the reviewer’s suggestion, introduction section has been rewritten and as per the suggested articles, literature has been added in the revised manuscript.
Round 2
Reviewer 1 Report
The authors considered most of the comments or adequately responded to the remarks contained in the review; therefore, the work may be approved for publication.
Author Response
Response to the Reviewer’s Comments:
The authors of this paper would like to sincerely thank the reviewer for their valuable comments made.

Reviewer 2 Report
Dear authors,
I’m satisfied with most of the modifications and improvements are done, however, there are some minor points that the authors didn’t consider or carelessly ignored which are addressed below in the attached file.

Author Response
Response (revision notes) to the Reviewer’s Comments:
The authors of this paper would like to sincerely thank the reviewer for their valuable comments/suggestions made. Please find the response for each comment in the following paragraphs.
- The abstract section can be incredibly improved if the authors can note some applications of the reinforced material.
Response: As per the reviewer’s suggestion, in the abstract application of reinforced material has been added, in the revised manuscript.
- Some minor English errors are still existing which authors are invited to correct carefully.
Response: As per the reviewer’s suggestion, the authors have checked and corrected several english errors and issues in the revised manuscript
- It is tried to enhance the introduction and literature review of the study but still, some gaps are necessary to be filled to encompass the comparison of various reinforced composite structures. These suggested articles addressed in the last review report had not been added to the literature review and introduction section:
- https://doi.org/10.1007/BF02782421
- https://doi.org/10.1140/epjp/i2019-12581-6
- https://doi.org/10.1007/s00366-020-00992-2
- https://doi.org/10.1007/s00366-020-01066-z
- https://doi.org/10.1007/s42405-019-00184-3
Response: As per the reviewer’s suggestion, the authors have added the suggested articles in the introduction part and have also cited the above articles in the revised manuscript.

Round 3
Reviewer 2 Report
Dear authors,
I checked the modifications applied to the revised manuscript and It deems satisfactory to me. I can say the manuscript can be accepted now.